# Responsible Job Design Based on the Internal Social Responsibility of Local Governments

**DOI:** 10.3390/ijerph17113994

**Published:** 2020-06-04

**Authors:** M. Isabel Sánchez-Hernández, Živilė Stankevičiūtė, Rafael Robina-Ramirez, Carlos Díaz-Caro

**Affiliations:** 1School of Economics Sciences and Business Administration, University of Extremadura, 06006 Badajoz, Spain; 2School of Economics and Business, Kaunas University of Technology, LT-44249 Kaunas, Lithuania; zivile.stankeviciute@ktu.lt; 3School of Business, Finance and Tourism, University of Extremadura, 10071 Caceres, Spain; rrobina@unex.es (R.R.-R.); carlosdc@unex.es (C.D.-C.)

**Keywords:** city council, communication, HRM, HPWP, internal social responsibility, job design

## Abstract

Sustainability needs the socially responsible orientation of public institutions, but not only externally. How civil servants and municipal employees consider what local governments do in relation to their internal social responsibility is the main question of this study. Through structural equation modelling and a sample of 294 employees in local governments in a Spanish region (Extremadura), it is demonstrated that responsible job position design, as well as good communication and team building, both have positive and significant effects on the personal identification of employees with the city council. That has been interpreted as the essence of internal social responsibility and the first step for building external social responsibility at the municipal level. Practical implications and future directions are discussed.

## 1. Introduction

Sustainable development is a global goal that implies meeting the needs of present generations without putting in danger the capacity of future generations to satisfy their own needs [1]. This means that sustainable organizations, private or public, have to seek the best conditions for life for everyone at present, and also for the coming generations [2,3]. In this context, the role adopted by the social responsibility (SR) of organizations is the obligation to reach maximum levels in their positive impacts on stakeholders, and to reduce as much as possible the negative ones [4].

In this context, SR is increasingly required not only of the private sector, but also of the public sector, whose main traditional functions are to provide accessible and qualitative service delivery, commercial activities, and stewardship of public assets to satisfy society (and not principally to obtain financial reward) [5]. At first glance, SR and sustainability are mainly linked to traditional public sector values, such as empathy, solidarity, fairness, integrity, or accountability [6,7]. In addition, SR is needed more than ever in the public sector, which has evolved from service delivery to co-production [8]. That means that government organizations are appealing to the intrinsic needs and social values of citizens. Under this new paradigm for understanding the public sector, it is expected going forward that available resources will be used effectively and efficiently, and new practices are gaining relevance. We refer to the emergence of collaborative government, public–private partnerships, or network governance [9,10,11,12].

In this complex context, in order to face the rapid evolving world, SR in local governments has to respond to new challenges that are emerging for public institutions. However, to the best of our knowledge, there are not yet substantial and relevant works devoted to the study and analysis of SR in public administrations and the relationship with their stakeholders.

Some related works approach citizens as a specific stakeholder in city councils, showing the external face of public SR. For instance, and under the theoretical framework developed by Osborne and Gaebler [13], citizens are considered as clients or customers. The focus of government policies and operations placing citizens as the central focus when designing public service delivery emphasizes concerns and needs. However, the movement, called “reinvented government administration”, has been questioned by other authors, such as Carroll [14], arguing that turning citizens into consumers marginalizes citizenship rather than activating participation. Other authors, such as Frederickson [15], have criticized the treatment of citizens as consumers, because citizens are really owners. Whatever it is, the SR of local governments under different approaches is undoubtedly a subject of debate.

In relation to the internal face of SR that considers employees as relevant stakeholders, it is true that human resource management (HRM) is increasing the attention of public management research. For instance, some studies have analyzed the HRM function in the public sector in relation to employee outcomes, such as job satisfaction or employee commitment [16,17], motivation [18,19,20], or public sector performance [21,22]. There is also a specific field of research devoted to comparing HRM in the private and public sectors [23,24,25,26]. In the same vein, an emergent body of academic literature exists on how the best HRM practices, called high-performance work practices (HPWP) [27,28], interrelate with each other and contribute to business performance [29,30], and some attempts exist to approach HPWP in public services [31], or even in local governments [32]. There are also studies linking HPWP to SR [33].

However, the reality is that there are not yet significant movements or relevant studies about internal SR in the public sector in general, and there are a lack of studies about the internal side of SR in local governments. To contribute to fill the gap, how civil servants and municipal employees consider what local governments do in relation to their internal social responsibility is the main question of this study. For that purpose, this paper deals with the development of a model to approach the SR of municipalities with civil servants and municipal employees.

After this introduction, Section 2 presents the theoretical background to develop hypotheses in Section 3. Later, Section 4 presents the method, Section 5 offers results, and Section 6 includes discussion and conclusions, including the limitations of the study and future lines of research.

## 2. Theoretical Background: Internal Social Responsibility in Local Governments

The stakeholder theory holds that besides shareholders, other groups are affected by an organization’s activity, and have to be also considered in managers´ decisions [34]. From this perspective, the collective efforts of the stakeholder network are at the core of value creation of any organization [35]. According to this theory, and following Sánchez-Hernández et al. [36], we can say that SR considers two different groups of stakeholders.

First, external SR considers the relationship of organizations with their communities. Organizations interact with their external stakeholders when they provide products or services by guaranteeing economic activity. That implies responsible actions, such as paying taxes, investing in the local economy, respecting human rights, and preserving the environment. Second and not less important, but still understudied, SR has an internal face that put the emphasis on employees. This internal consideration of SR adds a new perspective to the discussion about organizational management for sustainable development, by acknowledging that it is impossible to be socially responsible without a sound relationship with internal stakeholders. This mix of stakeholders and internal and external interacting with each other could be more or less sustainable when providing resources to the economic system, benefiting from the organization, and impacting the environment [37].

In recent years, HRM is starting to play a significant and helpful role in SR in organizations [38]. However, in contradiction, employees are still neglected stakeholders when conducting SR activities [39]. In the case of public organizations in general, and local governments in particular, the consideration of employees as internal stakeholders might be especially relevant, because these institutions have the duty to perform well, and to be good examples for different stakeholders, including their employees [40]. According to Mason and Simmons [41], employees expect SR values from their employing organization to be similar to other stakeholders, arguing that employees seek economic, functional, psychological, and ethical benefits from them. Consequently, if employers create, design, and offer stimulating work positions, some functional benefits will be obtained, and it will also be perceived as indicative of a socially responsible employer and a main driver of internal SR [42]. Thus, at the local government level, civil servants and municipal employees should be an important stakeholder to be considered. At this respect, it has been noted that civil servants and municipal employees have lower levels of motivation and satisfaction than private sector employees [43]. Boog and Cooper [44] also found that civil servants suffered more mental and physical illness. The main factors were intrinsic to the public job position, such as inequity regarding reward conditions and the feeling of having little control over their job and their institution.

Social identity theory is about individuals in groups, and the nature of social group processes. Since their origins, social identity has been seen as a social process [45]. The theory focuses on how identification works individually, how interactions are, and how these interactions impact on institutions [46]. It explains the role of self-conception, the associated cognitive processes, and the benefits for being part of a social group, including intergroup relations [47]. The central assumption of social identity theory is that group members will develop self-esteem, trust, and engagement, thus enhancing their external image [48].

Thus, civil servants, as individuals and municipalities, as organizations or social groups, are two sides of the same coin. Fostering social identification should improve the external image of municipalities.

Organizational identification refers to the degree to which someone defines himself (or herself) as having the same characteristics or traits that he (or she) believes define this organization [49]. Moreover, it is not a simply top-down shaping of the individual. According to Kuhn and Nelson [50], you are identified to your organization after an attaching process of shaping the organizational identities. Thus, organizational identification goes beyond job engagement [51] or organizational commitment [52]. In the context of local governments, we consider that the organizational identification of employees with their city council is the expected result of a successful internal SR. For instance, the expected public organization´s commitment to fairness and responsibility is reflected in designing jobs according to an equal opportunity philosophy. Discrimination between civil servants and municipal workers cannot exist, nor can any kind of discriminatory behavior at work because of sex or religion, among others [53].

To sum up, responsible local governments must be oriented to link HRM and SR development. For that purpose, human resource strategies must be redefined for considering new topics in public institutions, such as health prevention and safety at work, lifelong training programs, challenging systems of internal communication, and performance management systems, among others [54,55,56].

## 3. Hypotheses Development

### 3.1. The Importance of Designing Responsible Job Positions

Without underestimating other relevant issues related to the internal side of SR in any organization, and on the basis of responsible HRM, we want to highlight the relevance of a responsible job design (RJD). Unfortunately, the term RJD is not strongly embedded in academic literature at the moment, and neither is the term internal SR. Tangentially, some studies confront the good SR consideration of some companies when they have a bad reputation as employers. That has been the case with McDonald´s [57] and France Telecom [58], for instance. In these cases, where external SR is not balanced with internal SR, the organization is at risk of being accused of dishonesty and the SR should be considered as artificial.

Some authors have stated that job design essentially involves integrating job qualifications and responsibilities that are required to perform it [59].

In line with academic literature devoted to study ethics and the meaning of work [60,61,62], but also acknowledging the importance of traditional theories of work motivation and work design [63], RJD covers several dimensions, such as equal opportunities, well-paid work, salary in line with functions, recognition to the best suggestions and projects, autonomy at work, positive appreciation of the employee ideas by managers, and a responsible way of determining the worth of jobs, including a fair evaluation of employee performance. Along these lines, and considering that all existing measures are somehow incomplete, the scale developed by Morgeson and Humphrey [63], for instance, offers a set of indicators that are useful for approaching a socially responsible design.

Although there is no consensus among researchers about the particular human resource practices that are the most consistent with the sustainability approach, Thom and Zaugg [64] have argued that performance pay is one of the essential elements. This notwithstanding, well-paid work is one of the hot issues the public and private sectors are facing in the whole world. The importance of the mentioned issue is reflected in numerous ways. For instance, wages are included in the job quality index [65], or decent work concept, introduced by the International Labour Organisation (ILO), underlining the need for fair income. Well-paid work can reveal the level of care for employees provided by the employer, as our living quality standard is partly associated with the amount of money people earn. Moreover, a high level of pay can ensure that private and public sectors are able to attract and retain highly qualified employees [66]. Thus, the expected care of city councils for civil servants and municipality workers in terms of well-paid work indicates a responsible work design.

Salary in line with the employee’s functions emphasizes payment according to individual performance. Without underestimating different approaches concerning pay schemes in a public sector [67], we defend that local governments can elicit a certain level of performance from civil servants and municipal employees, by linking pay with functions and degree of responsibility. However, in this respect, Buurman et al. [68] found that only 45% of employees in the public sector in the Netherlands considered their salary to be adequate for the work they did. These findings do not fit the internal SR and opens the avenues for improving the job design.

Recognition of the best suggestions and projects is also an essential element of an RJD. According to the work psychodynamics theory, recognition is a reward expected by the subject that is largely symbolic in nature [69]. Mostly, recognition stems from a judgment made about various aspects. Thus, employee recognition can be expressed through four practices: personal recognition, recognition of work practices, recognition of job dedication, and recognition of results [69]. As stated by Saunderson [70], employee recognition in the public sector strongly correlates with good morale, loyalty, commitment, and satisfaction in the workplace. Moreover, recognition of employees serves as critical tool for their retaining. Bearing in mind that in the mentioned findings, it is supposed that the recognition of civil servants and municipal employees for the best suggestions and projects will increase the responsibility of any job design.

Morgesson and Humphrey [63] argue that nowadays, autonomy at work reflects the extent to which a job allows freedom, independence, and discretion to schedule work, make decisions, and choose the methods used to perform tasks. Freedom of civil servants and municipal employees in three domains, namely work scheduling, decision making, and work methods, strengthens the employee voice and participation [71].

Positive appreciation of employee ideas by managers is reflected in designing jobs according to the perceived manager support idea. Chughtai and Zafar [72] have emphasised that supervisors play an essential role in the employee–employer relationship. Results from previous studies give us an idea about the importance of support by the superiors of public employees [73]. Managers influence employee perceptions about the local government’s supportiveness. Moreover, managers have an impact on the extent to which employees can expect that the organization will look after their interests, i.e., behave in a socially responsible way.

Socially responsible organizations must attend to the fit of employees with the values of the organization [74]. Consequently, job evaluation, as the judgment about value or worth of the jobs, might be a feature of responsible work places. A responsible job evaluation begins with the analysis of work to determine its characteristics and requirements, and then follows with fair employee performance management. Employee performance evaluation is related to goal setting and providing feedback. As stated by Verbeeten [75], by setting clear goals and “measuring whether they are achieved, organisations reduce and eliminate ambiguity and confusion about objectives, and gain coherence and focus in pursuit of their mission”. The expected local government commitment to the fair evaluation of employee performance is reflected in designing jobs in line with the organisational justice philosophy. It is supposed that procedural, interactional, interpersonal, and informational justice dominate in a responsible work design [76].

A responsible design of job positions in public institutions is expected to be linked to higher levels of organizational identification of employees with their city council. The better the jobs are designed, the higher the level of identification will be. In the same vein, the better the communication and team building efforts are, the higher the personal identification of employees will be with the city council where they work. Taking into consideration all the above, the following hypothesis emerges:

**Hypothesis** **(H1).**
*The responsible design of jobs in local governments is positively related to the employee´s identification with the city council.*


### 3.2. The Role of Communication and Team Building

RJD is not enough for developing an effective internal SR. Communication and team building play an essential role in allowing things happen. At its simplest, communication could be defined as social interaction through messages [77,78]. In a huge part of literature, the significant role of organizational communication (internal and external) is acknowledged as it persists throughout the lifespan of the organisation [79].

From the perspective of the internal side of SR in any organization, internal communication is a matter of interest following evidence that the outcomes of effective internal communication are beneficial for both sides: the whole organization and employees as stakeholders. For instance, Welch and Jackson [80] stated that effective internal communication influences employee productivity and the behaviour of the organization. Thomas et al. [81] argued that communication plays an essential role in the development of trust within any organisation, and Martin-García and Conci [82] highlighted that informal participation and communication are considered as frequent HPWP in countries such as Spain. Going further, we can affirm that effective internal communication enables the success of teambuilding and teamwork [83]. Recently, teamwork has been getting more interest, assuming that integrated teamwork is more important than individual work in reaching organizational goals [84].

Communication and teambuilding are very important in public institutions [85,86]. Effective internal communication is crucial for success, as it affects the ability of managers to engage employees and achieve objectives [87]. Moreover, internal communication is designed to promote commitment to the organization, a sense of belonging to it, awareness of its changing environment, and understanding of its evolving aims [80]. Effective communication supports teamwork, and team relations appear to be an important motivator for public employees [63]. Communication pathways and channels or teambuilding are supposed to have positive relationships with job design and the personal identification of employees. Finally, and taking into consideration all the above, the following hypotheses emerge:

**Hypothesis** **(H2).**
*The responsible design of jobs in local governments is positively related to communication and team building developments.*


**Hypothesis** **(H3).**
*Communication and team building efforts are positively related to the employees’ identification with the city council.*


## 4. Method

### 4.1. Tools

Structural equation modelling (SEM) was considered adequate to test the hypotheses of this study. According to Fornell and Larcker [88], SEM shows cause–effect relationships between constructs. SEM modeling distinguishes two different approaches, assuming that one method is not superior to the other. Sometimes SEM is equivalent to carrying out covariance-based analyses, using software like LISREL for instance. Our approach is different. We used partial least squares (PLS), which is a causal modeling approach aimed at maximizing the explained variance of the dependent latent constructs [89,90].

PLS-SEM has as the main characteristic of having higher statistical power, and this is quite useful for research that examines still-developing theory [91,92], as it is in our case. In addition, we have used the original PLS algorithm, because the aim is definitely exploratory but somehow predictive, and in both cases it is recommended by authors like Sarsted et al. [93] to use composites instead of common factor.

This methodological tool has the capacity to confront theory and data through a system of multiple regressions between constructs. In our study, there are three latent variables to be analyzed: RJD, communication and team building, and the employees´ identification with the city council. The three constructs are latent variables, not directly observable ones that need indicators (observable variables) to be approached, as will be shown later.

In this study, we have used the software Smart-PLS (Partial Least Squares), developed by Ringle et al. [94]. The estimation of parameters is based on minimizing the residual variances of the endogenous variables and maximizing the explained variance (*R*^2^) of the dependent variables.

### 4.2. Measurement Scales

Previous literature review has served the purpose of defining a selection of items approaching RJD, communication, and team-building. The organizational identification of civil servants and municipal employees with the city council is in line with the SR and sustainability framework. Social identity theory argues that people classify themselves as belonging to various social categories according to interests, skills, age, etc. [95]. Having an organizational context in mind, most research deals with employee identification with a particular team, workgroup, department, occupation, and organisation [96]—in this case, the identification with the city council. In this work, the construct has been approached from the field of marketing, following Mael and Ashforth [97]. To sum up, and supported by the literature review, the selected indicators for this study are shown in Table 1, as follows.

### 4.3. Procedure and Sample

An empirical study was carried out to contrast the hypotheses derived from the model and validate them for the region under study. It is important to note the difficulty for accessing employees, as official procedures were numerous and red tape was a barrier for field work. That was the reason for using a nonprobability sampling, in which employees were sampled because they were accessible sources of data for researchers.

In total, 294 participants, civil servants, or municipal employees from an initial population of 1200 employees from 12 different city councils in the Autonomous Region of Extremadura answered a questionnaire, with five-point Likert questions measuring the variables with the selected indicators. The technical data sheet is shown in Table 2. No distinction was made between civil servants and municipal employees, because their labor situations were not linked to specific roles or functions.

### 4.4. Model

The theoretical model developed (Figure 1) represents the expected relationship between RJD, on the basis of the internal SR of the city council, and the expected positive impact on the organizational identification of civil servants and municipal employees with the city council. In a local government, internal SR should start by designing responsible jobs for employees to guarantee the process of shaping the desired responsible city council identity to which public employees attach. The model also considers the mediating role of communication and teamwork in enabling the process.

## 5. Results

Firstly, we present results from the measurement model (the “inner model”). For evaluating the measurement model, we calculated reliability, which is an attribute considering whether the process is stable and consistent [98]. It is the first step in reflective measurement model assessment, and involves examining the indicator loadings. Loadings above 0.7 are recommended, as they indicate that the construct explains more than 50% of the indicator’s variance, thus providing acceptable item reliability [92]. This attribute was assessed by examining simple correlations of the measures with their respective latent variables. Considering the exploratory nature of the study, a value with 0.64 was accepted. Table 3 shows the loadings of each observed variable, demonstrating that depuration of items was not needed.

Table 4 analyzes and confirms that our constructs were properly measured by the indicators. The Cronbach’s alpha coefficient was used as an index of reliability of the latent variables. Composite reliability was also calculated. Validity was also considered, which is an attribute measuring what one really wanted to measure [98]. The convergent validity of each construct was evaluated through the average variance expected (AVE) (accepted when >0.5). The discriminant validity of constructs was verified using the Fornell–Larcker criterion [99] when examining whether the square root of the AVE value of each item was above the correlations with the other latent variables. In addition, according to Henseler et al. [100], a test was conducted to demonstrate whether another technique better detects the potential lack of discriminant validity: the heterotrait–monotrait (HTMT) relationship test. Discriminant validity was also confirmed, as HTMT ratios for each pair of factors were <0.90 [100].

Secondly, we carry out the analysis of the structural model (“outer model”) for verifying whether the model considers the proposed relationships between the constructs. For this purpose, the path coefficients (β) were examined, along with their respective levels of significance. Although PLS estimators lack the parameter precision of maximum-likelihood estimation in achieving optimal prediction, we provided values for *R*^2^ (the explained variance of the endogenous variables), and the root mean square residual (SRMR), even acknowledging that it is only necessary for confirmatory composite analysis [101]. The overall fit of the model was evaluated using the standardised root mean square residual indicator (SRMR). Hu and Bentler [102] defined SRMR as the root mean square difference between the correlations observed and the correlations implicit in the model. A cut-off value of 0.08 for SRMR is considered the most appropriate in PLS [103]. In this study, the SRMR was 0.078, which means that the model fits the empirical data [98].

Taking into account that the principal usefulness of the PLS methodology is to predict potential cause–effect impacts between constructs into a model, the goodness of a model is mainly determined by the strength of each structural path, and is analyzed using the value of *R*^2^ (explained variance) for the dependent latent variables. The values of *R*^2^ obtained for research has led to the following conclusions: 0.67 is considered to be good, 0.33 is medium, and 0.19 is weak [104]. The result obtained for the principal dependent variable in the model organizational identification (OI) was *R*^2^ = 0.47 The evidence, therefore, shows that the presented model has a substantial predictive capacity. This explains why both the RJD and communication and team-building (CTB) contribute decisively to a successful OI.

The predictive relevance of the model was also studied through the blindfolding technique, and it demonstrates that the model has predictive capacity. This technique consists of omitting part of the data of a given construct during the estimation of the parameters, and then trying to estimate what was excluded from the estimated parameters [104]. According to the results obtained (*Q*^2^ (Predictive relevance measure in PLS) for CTB = 0.340 and *Q*^2^ for RJD = 0.251) all endogenous constructs fulfil *Q*^2^ > 0. Following the Stone–Geisser (*Q*²) test [105,106], the values are 0.02, 0.15, and 0.35, indicating small, medium, and high predictive relevance, respectively. As a result, both the CTB and RJD constructs have medium–high predictive relevance.

Complementary to htis, PLS_predict_ was calculated following Hair et al.’s procedure [107] (Table 5). Setting *k* (*k* is the default number of equally sized subsets of data) = 10, *Q*^2^_predict_ values >0 indicate that the model outperforms the most naïve benchmark. Comparing the root mean squared error values (RMSE) with the linear regression model value (LM) of each indicator, we found that the majority of the indicators presented RMSE_PLS_ < RMSE_LM_^,^ corroborating the medium–high predictive power of the model.

Finally, the bootstrapping procedure was used for hypotheses testing (Table 6). Bootstrapping is a non-parametric resampling procedure that assesses the variability of a statistic by examining the variability of the sample data [108]. The results obtained allow the accepting all the hypotheses, since there were no statistically significant differences in the relationships between the variables in our model (value of *p* > 0.05).

## 6. Discussion

Sustainable development (people, planet, and profit) must be addressed from private organizations, but also from public entities, and city councils are an important part of those. This work contributes to both the theoretical and empirical literature within SR in the public sector. In terms of theory, the paper contributes to the literature on SR, revealing the relevance of the internal side related to HRM in general and HPWP in particular. Moreover, the paper adds value to the state-of-the-art of literature approaching the internal SR of municipalities and including the perception of civil servants and municipal employees, using a new set of variables that are specifically suited to the public sector at the local level.

Considering the lack of previous studies devoted to this topic and the exploratory nature of this pilot study to approach the internal responsibility of city councils, we have put the attention on RJD according to the civil servants and municipal employees´ perceptions and the linkages to their social identification. The novelty of the article is twofold: first, the conceptualization of the term RJD, and second, the empirical exploration of the latent variables of this construct under the social identity theory.

## 7. Conclusions

The emphasis on RJD as the basis of internal SR in the context of city councils provides a novel and specific context for the analysis of the emerging issue of HPWP in the public sector, and shifts the emphasis away from the traditional general studies comparing private and public HRM [23,24,25,26]. This article simultaneously addressed the application of HPWP in the public sector for engaging employees to their institutions. Thus, we can conclude that the adoption of RJD as an HPWP in city councils provides a mechanism whereby local governments can consider their internal SR as strategic for sustainable improvement and performance [27].

The paper provides also some practical implications for public policy at the local level that could be considered future lines of research. First, it would be valuable to further examine how civil servants and municipal employees construct their identification with the city council, and how this impacts their performance, and eventually, the citizens´ satisfaction. Secondly, and directly related, an area that deserves practical attention is exploring the opportunities that exist for linking internal and external SR in public institutions. Broadly speaking, there is a deficit of connection between HRM/HPWP and external public policies for enhancing sustainability and related issues.

Thus, in line with the scarce previous works [31,32,33], and taking into consideration the results obtained from the empirical study, we can conclude the following: high-performance-work city councils could be described as local governments with high levels of internal RS that are putting the emphasis on RJD, internal communication, and team development to guarantee engaged and empowered civil servants and municipal workforces, who are very identified with the council and offer high-quality goods and services.

The paper has some limitations that will have to be overcome in future research—primarily, the use of data from a region in a single country. This study will be extended to other regions in Spain, but also to other regions in other countries, in order to get insights from different contexts and to refine the model for a better and more generalizable understanding of the internal side of SR in local governments.

Another relative limitation is the fact that the design of the study was not longitudinal. SEM-PLS is specially recommended for empirically testing theoretical cause–effect processes [91,92,93], and the results obtained from the new function PLS_predict_ [108] satisfactorily confirm the predictor power of the model tested.

However, new studies are needed in this line of research. The purpose here was only exploratory, and that was the reason to use PLS and composites, as recommended by Sarstedt et al. [93], acknowledging that it was not clear whether SR, RJD, and OI were composites or common factors. The good results obtained both in the measurement and structural model promote continuing in this line of research and looking for more explicative and complex models in the near future.

## Figures and Tables

**Figure 1 ijerph-17-03994-f001:**
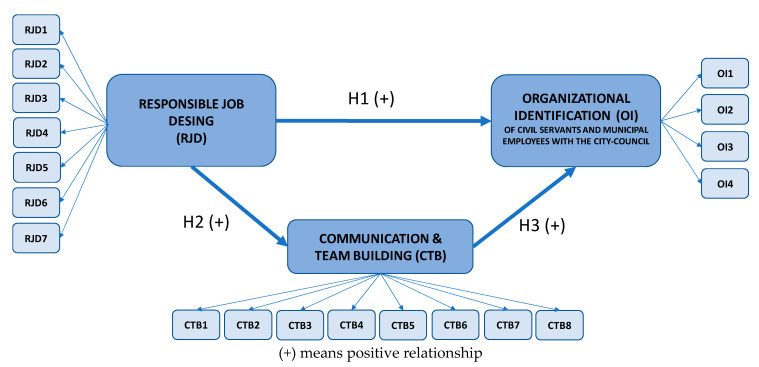
Theoretical model.

**Table 1 ijerph-17-03994-t001:** Measurement tool.

Constructs, Indicators And Main Sources	
**• Responsible Job Design (RJD)**	
Equal Opportunities Policy (RJD1)	Morgeson And Humphrey [64]
Well-Paid Work Compared to other Public Administrations (RJD2)	
Salary in Line with the Functions and the Degree of Responsibility Assumed (RJD3)	
Recognition of the Best Suggestions and Projects (RJD4)	
Autonomy at Work (RJD5)	
Ideas Taken into Account By Superiors (RJD6)	
Jobs Well-Valued in the Organization (RJD7)	
**• Communication and Team Building (CTB)**	
Management Group Availability (CTB1)	Thomas Et Al. [82]
Good Communication Top–Down (CTB2)	
Good Communication Down–Top (CTB3)	
An E-Mail Should be Enough for Controversial Subjects (CTB4)	
Knowledge of the Reports Published by the City Council (CTB5)	
Adequate Internal Communication Channels (CTB6)	
Team Work is Encouraged by the Management Group (CTB7)	
Adequate Degree of Demand towards Employees (CTB8)	
**• Organizational Identification (OI)**	
Feeling of Pride for Belonging to the City Council (OI1)	Mael And Ashforth [98]
Feeling of Integration into the City Council (OI2)	
In Accordance with the Service that the City Council Provides to the Citizens (OI3)	
Identification with the Mission, Vision, and Values of the City Council (OI4)	

**Table 2 ijerph-17-03994-t002:** Technical data sheet.

Population and Geographical Scope	1200 Employees from 12 City Councils in Extremadura (Region in Spain)
Method of Information Collection	Personal contact
Sample	294 employees (civil servants and municipal employees)
Measurement Error	5%
Confidence Interval	95% *z* = 1.96; *p* = q = 0.5
Sampling Method	Convenience sampling
Average Duration of the Interview	10 min

*z* is the value of the distribution function. For *95*% *confidence* is *z* = *1.96*. *p* = q = 0.5 (Assuming equal proportions)

**Table 3 ijerph-17-03994-t003:** Loadings.

Constructs	Items	Loading
**RJD**	RJD1	0.701
	RJD2	0.774
	RJD3	0.725
	RJD4	0.826
	RDJ5	0.640
	RDJ6	0.725
	RDJ7	0.825
**CTB**	CTB1	0.788
	CTB2	0.885
	CTB3	0.831
	CTB4	0.740
	CTB5	0.710
	CTB6	0.822
	CTB7	0.796
	CTB8	0.788
**OI**	OI1	0.711
	OI2	0.748
	OI3	0.837
	OI4	0.850

**Table 4 ijerph-17-03994-t004:** Inner model results.

Constructs	Cronbach´s Alpha	rho_A	Composite Reliability	AVE	Fornell–Larcker Criterion			Heterotrait–Monotrait Ratio		
					RJD	CTB	OI	RJD	CTB	OI
**RJD**	0.867	0.876	0.898	0.559	0.748					
**CTB**	0.917	0.921	0.933	0.635	0.785	0.797		0.869		
**OI**	0.800	0.825	0.867	0.622	0.570	0.676	0.789	0.666	0.765	

rho_A reliability measure for PLS; AVE reliability measure for PLS.

**Table 5 ijerph-17-03994-t005:** PLS_prediction_ summary.

Indicators	PLS		LM	
	RMSE	*Q* ^2^	RMSE	*Q* ^2^
CTB1	0.977	0.267	0.978	0.265
CTB2 *	0.868	0.435	0.879	0.421
CTB3 *	0.899	0.402	0.905	0.396
CTB4	0.902	0.344	0.882	0.373
CTB5 *	1.018	0.253	1.038	0.224
CTB6 *	0.853	0.466	0.863	0.455
CTB7	0.875	0.478	0.867	0.488
CTB8	0.861	0.428	0.860	0.429
OI1 *	0.698	0.147	0.707	0.125
OI2 *	0.757	0.147	0.762	0.135
OI3	0.941	0.228	0.936	0.237
OI4 *	0.856	0.251	0.865	0.236

Note: * means RMSE_PLS_ < RMSE_LM._
*Q*^2^**:** Predictive relevance measure in PLS

**Table 6 ijerph-17-03994-t006:** Hypotheses testing.

Hypothesis: A → B	Original Path Coefficient (β)	Mean of Sub-Sample Path Coefficient	Lower IC (25%)	Higher IC (97.5%)	Standard Deviation	*T*-Statistic	*p*-Value
Direct Effects							
H1: RJD → OI	0.570	0.573	0.481	0.654	0.044	13.027	0.000 ***
H2: RJD → CTB	0.785	0.785	0.735	0.829	0.024	32.496	0.000 ***
H3: CTB → OI	0.595	0.595	0.437	0.733	0.077	7.782	0.000 ***
Indirect Effects							
RJD → CTB → OI	0.468	0.464	0.343	0.583	0.062	7.544	0.000 ***

*** *p* < 0.001 (based on a Student’s two-tailed test, *t*_(499)_); *t*_(0.05;499)_ = 1.96). → direct relationship.

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
