# Peer review of "Responsible Job Design Based on the Internal Social Responsibility of Local Governments"

_ijerph, 2020, doi:10.3390/ijerph17113994_

Round 1

Reviewer 1 Report

Introduction, the statement and the hypothesis of the article are very interesting. I have comments about the statistics of the model.

To demonstrate the validity of a Structural Equation Model, there are some reliability indices. The most common are: degrees of freedom, Chi-Square, GFI, AGFI, PGFI, RMSEA, CFI, NFI, and IFI. I have not used the software Smart PLS and I don´t know if it gives these results. I see in lines 276-283 information about the "outer model" showing R2 and SRMR. The model can´t be approved without the fit indices. The model can be optimized considering these indices.

The SEM was structured with three constructs (latent variables): RJD (with 7 observed variables), CTB (with 8 observed variables), and OI (with four variables). It is not clear if all the observed variables remained in the model after the first evaluation. For example, if one observed variable is eliminated from the model due to its non-significance, the fit indices will change. In the article, this analysis is not clear.

Also, the factor loading of each observed variable is not presented. I suggest to include a table or figure where all factor loadings and indices of the model be presented.

I suggest to include a table where direct and indirect effects on the model are presented. For example, it is demonstrated that CTB has a positive and direct effect (0.595) on the OI, but RJD also presented both, an direct (0.570) and indirect effect (through CTB) on OI.

Finally, please review some minor issues about grammar writing.

Reviewer 2 Report

The paper take into  account the relevance of Responsabile job design, Communications and team building and Organizational identification: three  constructs of wellbeing at work and efficiency.

The authours tested cause-effect relationships among these three constructs and their predictive capacity. Unfortunately the design of the study is not longitudinal, perhaps it is better discuss this point in the limit section, even if the authours declared SEM able to test cause-effect.

The readers could be not aware of the differences existing between civil servants and municipal employers as well as their specific roles in city council. Are all they together an homogeneus population? Some clarification on this point could be appropriate.

Reviewer 3 Report

Dear Authors,

Thanks for the opportunity to review the manuscript titled: “Responsible job design on the basis of the internal social responsibility of local governments".

The manuscript addresses an issue of the social responsible orientation of public institutions. The topic itself is interesting, and within the interest of International Journal of Environmental Research and Public Health

The paper has several shortcomings that need to be addressed before it will be ready for publication.

In my opinion, there are some weaknesses that need to be amended:

  1. I would recommend describing the organization of a paper in the introduction section.

  1. I would suggest Authors elaborate on hypotheses development. Developing each  hypothesis separately will improve the quality of the manuscript.

  1. The Authors mention Social identity theory in the measurement section. In my opinion, pointing it out in the hypothesis development regarding RJD would definitely provide a broader perspective of the conceptualization.

  1. The more specific description of measures used in this research is needed. How has it been developed? Authors claim that following the literature review they have chosen certain indicator (line 239)- I would recommend to include the references used for each item.

  1. My biggest concern refers to sampling.

  • The Authors mention that they use random sampling. However, they don’t specify the sampling rule adhere in-order to select the sample.

  • In order to estimate the measurement error and internal the population size is needed. Without it , it is hard to estimate the needed sample size and desired precision.

  • Furthermore, there is not sth as “trust level” – (table 2) I believe Authors refer to confidence interval”

  • I assume it is a cluster random sampling Philip Sedgwick, Cluster sampling, January 2014BMJ (online) 348(jan31 2):g1215-g1215 DOI: 10.1136/bmj.g1215?
  1. There are some language mistakes in the manuscript. I would recommend proofreading a manuscript by a native speaker.
  2.  

7.1. Table 3 please replace Alfa de Cronbach with Cronbach's alpha

7.2. the Authors haven’t conducted CFA, which is a mandatory statistic procedure in SEM. I would recommend adding CFA to the SEM model .

Hair, J. F., Black, W. C., Babin, B. J., & Anderson, R. E. (2010). Multivariate data analysis (7th ed.). Prentice Hall, Englewood Cliffs.

7.3. Line 289 – authors write R2 = 46.1%.. I believe it should be R2=0.47  

  1. Conclusion : Authors should extend this section by focusing on a discussion.

  1. Another significant concern is that the bibliographical references. The majority of literature used in this manuscript is are out-of-date (most bibliographical sources are older than 10 years). Therefore, it is necessary to update and improve the bibliographical references.

In my opinion, the manuscript needs a major revision. In conclusion, this paper is good, but some improvements are necessary for the final publication.

Reviewer 4 Report

Thank you for submitting your article to IJoERPH,

My overall evaluation of this work is positive, however I have a number of comments that can help them to improve the paper.

I would encourage the authors to revise  the main research questions, by moving away from the focus on _how local governments consider  SR with civil servants and employees_  towards  how the servants and employees consider what the governments do. This focus should be revised throughout the whole narrative of the paper.

In the introduction section I do not find it particularly relevant to introduce the literature on external SR.

It is advisable to provide more argumentation, in the paper, on the employment of the concept of RJD. The concept  of "design" may not  necesarilly be associated with the actual responsible practices employed in a workplace environment. How strongly embededd this concept is in the literature?

And I would expect to see what is the novelty of this empirical contribution?

Your reviewer

Round 2

Reviewer 1 Report

The revised version has been improved considering my comments. Congratulations for this new version.

Reviewer 3 Report

Dear Authors,

Thank you for including all the remarks and suggestion.

Your manuscript is now ready to be published.

Best Regards.